# Oxidative Stress Modulation by Cameroonian Spice Extracts in HepG2 Cells: Involvement of Nrf2 and Improvement of Glucose Uptake

**DOI:** 10.3390/metabo10050182

**Published:** 2020-05-01

**Authors:** Achille Parfait Atchan Nwakiban, Stefania Cicolari, Stefano Piazza, Fabrizio Gelmini, Enrico Sangiovanni, Giulia Martinelli, Lorenzo Bossi, Eugénie Carpentier-Maguire, Armelle Deutou Tchamgoue, Gabriel A. Agbor, Jules-Roger Kuiaté, Giangiacomo Beretta, Mario Dell’Agli, Paolo Magni

**Affiliations:** 1Department of Biochemistry, Faculty of Science, University of Dschang, P.O. Box 96 Dschang, Cameroon; achilestyle@yahoo.fr (A.P.A.N.); jrkuiate@yahoo.com (J.-R.K.); 2Department of Pharmacological and Biomolecular Sciences, Università degli Studi di Milano, Via G. Balzaretti 9, 20133 Milano, Italy; stefania.cicolari@unimi.it (S.C.); stefano.piazza@unimi.it (S.P.); enrico.sangiovanni@unimi.it (E.S.); giulia.martinelli@unimi.it (G.M.); lorenzo.bossi@unimi.it (L.B.); 3Department of Environmental Science and Policy, Università degli Studi di Milano, via G. Celoria 2, 20133 Milano, Italy; fabrizio.gelmini@unimi.it (F.G.); giangiacomo.beretta@unimi.it (G.B.); 4Department of Science and Technology, University of Lille, Rue de Lille, 59160 Lille, France; eugenie.carpentiermaguire@gmail.com; 5Centre for Research on Medicinal Plants and Traditional Medicine, Institute of Medical Research and Medicinal Plants Studies (IMPM), Yaoundé 4124, Cameroon; armelle_d2002@yahoo.fr (A.D.T.); agogae@yahoo.fr (G.A.A.); 6IRCCS MultiMedica, Sesto San Giovanni, Via Milanese, 300, 20099 Sesto San Giovanni Milan, Italy

**Keywords:** Cameroonian spice extracts, oxidative stress, HepG2 cells, HPLC-UV-DAD, Nrf2, glucose uptake

## Abstract

Oxidative stress plays a relevant role in the progression of chronic conditions, including cardiometabolic diseases. Several Cameroonian plants, including spices, are traditionally used as herbal medicines for the treatment of diseases where oxidative stress contributes to insulin resistance, like type 2 diabetes mellitus. This study evaluated the antioxidant capacity and the effects on oxidative-stress-induced impairment of glucose uptake of 11 Cameroonian spice extracts. H_2_O_2_-induced reactive oxygen species (ROS) production by human HepG2 cells was significantly reduced by 8/11 extracts. The most effective extracts, *Xylopia parviflora*, *Echinops giganteus*, and *Dichrostachys glomerata*, showed a concentration-dependent ROS-scavenging activity, which involved Nrf2 translocation into the nucleus. *Xylopia parviflora*, *Tetrapleura tetraptera*, *Dichrostachys glomerata*, *Aframomum melegueta*, and *Aframomum citratum* extracts showed the highest antioxidant capacity, according to oxygen radical absorbance capacity (ORAC) (2.52–88 μM Trolox Eq/g of extract), ferric-reducing antioxidant power (FRAP) (40.23–233.84 mg gallic acid Eq/g of extract), and total phenol (8.96–32.96% mg gallic acid Eq/g of extract) assays. In HepG2 cells, glucose uptake was stimulated by 4/11 extracts, similarly to insulin and metformin. H_2_O_2_-induced oxidative stress reduced glucose uptake, which was rescued by pretreatment with *Xylopia aethiopica*, *Xylopia parviflora*, *Scorodophloeus zenkeri*, *Monodora myristica*, and *Dichrostachys glomerata* extracts. The ROS-scavenging ability of the spice extracts may reside in some secondary metabolites observed by phytochemical profiling (reverse-phase high-performance liquid chromatography coupled to a diode array detector (HPLC-UV-DAD)). Further studies are needed to better clarify their biological activities and potential use to control oxidative stress and promote insulin sensitivity.

## 1. Introduction

Oxidative stress is recognized as an important pathophysiological factor leading to molecular and cellular tissue damage [1]. It plays a relevant role in the process of aging and in the pathogenesis of cardiometabolic disorders, including obesity, type 2 diabetes mellitus (T2DM), metabolic syndrome, and insulin resistance [2]. It has been well documented that increased reactive oxygen species (ROS) production and derangement of endogenous antioxidant defense mechanisms play a pathophysiological role in hyperglycemia-induced cellular and tissue damage [3]. Moreover, in T2DM, several studies have shown a strong association between increased intake of exogenous antioxidants, from both pharmacological (i.e., metformin) [4] and nutritional sources, and reduction of oxidative-stress-associated biomarkers [5]. Therefore, the ingestion of dietary antioxidants from foods, herbs, and spices is the most recommended approach to counteract macromolecular damage by oxidative stress, and the search for natural compounds able to contrast oxidative stress is of major interest. In this regard, plant-derived products are widely used in Africa for the treatment of many ailments and traditionally constitute the first line of health support for about 70–80% of the population [6]. The reasons for such massive use of traditional herbal medicines include cultural and economic aspects, such as the high cost and logistical difficulties associated with some pharmaceutical approaches [7]. Although the efficacy of some traditional medicines for the treatment of several ailments has been demonstrated, questions about their safety of use and toxicity have also emerged, highlighting the need for a detailed characterization of the efficacy/safety balance [8].

Interestingly, some spices commonly used in Cameroon for both nutritional and medicinal purposes have been reported to contain high levels of polyphenols with ROS-scavenging properties, and thus potentially to be able to decrease the adverse effects of ROS [9,10]. Based on these observations, this study investigated the antioxidant potential of 11 selected spice extracts from Cameroon and evaluated some associated molecular mechanisms and their ability to modulate glucose uptake in a cell-based model system after exposure to oxidative stress. For this purpose, we selected the human hepatoma HepG2 cell line, which is a widely used experimental model of human liver cells, often utilized in studies on oxidative stress and glucose uptake/insulin resistance [11].

## 2. Results

### 2.1. Preparation of Spice Extracts

Material from the 11 plants (nutritional spices) was extracted according to the reported protocol [12]. The extraction yield from fruits was 6.4–49.2%, from seeds was 7.1–32.7%, and from root samples was 13.1% (Table 1). Most of the extracts were in solid form (powders or crystals), except for *Monodora myristica*, which yielded an oily liquid form. The extraction yield was 7.1–32.7% for seeds, 6.4–49.2% for fruits, and 13.1% for the root samples (Table 1).

### 2.2. Effects of Spice Extracts On HepG2 Cell Viability and Morphology

The cytotoxicity of spice extracts was evaluated via MTS [3-(4,5-dimethylthiazol-2-yl)-5-(3-carboxymethoxyphenyl)-2-(4-sulfophenyl)-2H-tetrazolium] assay. HepG2 cells were treated with increasing concentrations (0–100 μg/mL) of extracts for 24 and 48 h. Viability threshold was set at 80% [13]. Toxicity at 50 and 100 μg/mL was observed for 7/11 extracts in HepG2 cells (Figure 1; Appendix A). Since extract concentrations ≤ 25 μg/mL proved to be non-toxic for all spice extracts, we used this or lower concentrations for further experiments. As an additional viability evaluation, we also examined cell morphology after treatment with 1 and 100 μg/mL spice extracts (Appendix A). No morphological changes were observed for any spice extracts at 1 μg/mL (data not shown). At 100 μg/mL, after 48 h, most spice extracts were toxic, and severe morphological changes leading to cell death, including rounding and shrinking of the cells, disintegration of the membranes, and cytoplasmic aggregation were observed in HepG2 cells (Appendix A). By contrast, no morphological changes were observed in cells treated with *Afrostyrax lepidophyllus*, *Monodora myristica*, or *Echinops giganteus* (data not shown).

### 2.3. Effect of Spice Extracts on ROS Production in HepG2 Cells

The protective effect of antioxidants includes their ability to increase the cellular antioxidant defense mechanisms as well as to reduce the level of intracellular ROS generation. To test this latter effect, intracellular ROS generation was evaluated in HepG2 cells using the 5-(and-6)-Chloromethyl-2’,7’-dichlorodihydrofluorescein diacetate (CM-H2DCFDA) assay. A 1 h treatment with 500 µM hydrogen peroxide (H_2_O_2_) resulted in a two-fold increase of intracellular ROS levels and, as expected, co-treatment with 500 µM Trolox, a potent antioxidant, was able to substantially prevent such an increase (Table 2). After cell pretreatment for 24 h with spice extracts (all at 10 µg/mL), ROS production induced by H_2_O_2_ was reduced to a different extent by all spices except *Scorodophloeus zenkeri*, *Monodora myristica*, and *Afrostyrax lepidophyllus* (Table 2). Furthermore, some of the most effective extracts, like *Xylopia parviflora*, *Echinops giganteus*, and *Dichrostachys glomerata*, were found to exert their ROS-scavenging activity in a concentration-dependent fashion (Figure 2).

### 2.4. Evaluation of Antioxidant Ability of Spice Extracts

Additional information on the antioxidant properties of Cameroonian spice extracts was obtained through different biochemical assays, including phenol content quantification, oxygen radical absorbance capacity (ORAC), and ferric-reducing antioxidant power (FRAP) assays. Spice extracts, tested at the final concentration of 1 μg/mL, were found to contain different amounts of phenols (Figure 3A). The values ranged from 0.9% to 34.4% of phenols, expressed as gallic acid equivalents (GAE) per g of extract. The highest phenol content (> 18% GAE/g of sample) was found in extracts from *Xylopia parviflora* (34.4%), *Dichrostachys glomerata* (27.0%), *Aframomum melegueta* (22.3%), and *Aframomum citratum* (18.5%), whereas the lowest content was present in *Monodora myristica* (0.9%) and *Scorodophloeus zenkeri* (3.9%). The antioxidant activity of spice extracts was assessed by ORAC (testing extracts at 0.1 μg/mL) and FRAP (testing extracts at 37 μg/mL) assays. Spice extracts of *Xylopia parviflora*, *Aframomum melegueta*, *Aframomum citratum*, *Echinops giganteus*, and *Dichrostachys glomerata* showed a relevant activity, with values of 8.7 ± 3.6; 11.9 ± 3.2; 7.3 ± 0.1; 7.0 ± 2.1; 4.3 ± 0.9 μM Trolox equivalent g of extract (ORAC assay) and 233.8 ± 4.9; 138.6 ± 4.8; 130.2 ± 5.2; 184.3 ± 0.7 mg equivalent gallic acid/g of extract (FRAP), respectively (Figure 3B, 3C). Interestingly, the combined analysis of the results obtained with these assays showed that they were significantly (*p* < 0.01) correlated (R^2^ = 0.618 between ORAC and phenol content; R^2^ = 0.678 between ORAC and FRAP; R^2^ = 0.979 between phenol content and FRAP), thus suggesting that phenols may be responsible for the radical-scavenging effects exerted by these spice extracts.

### 2.5. Nuclear Translocation of Nrf2

Cells counteract the negative effects of ROS through several molecular mechanisms, such as the nuclear factor E2-related factor 2 (Nrf2) [14]. Therefore, the effect of two of the most potent ROS scavenging spices reported above, *Dichrostachys glomerata* and *Xylopia parviflora,* on Nrf2 translocation into the nucleus was assessed. According to immunofluorescence analysis, H_2_O_2_ treatment increased nuclear Nrf2 abundance. *Dichrostachys glomerata* treatment resulted in a reduced Nrf2 immunofluorescence in the nucleus compared to controls, but was unable to counteract the effect of H_2_O_2_ (Figure 4). Compared to controls, treatment with *Xylopia parviflora* elicited an increase of nuclear Nrf2 levels, which was not modified in the presence of H_2_O_2_ (Figure 4).

### 2.6. Fingerprinting of Spice Extracts

The spice extract compositional profiles were evaluated via reverse-phase high-performance liquid chromatography coupled to a diode array detector (HPLC-UV-DAD) (Figure 5). The chromatogram of *Xylopia aethiopica* fruit hydroalcoholic extract was dominated by three peaks generated by substances with UV absorption maxima in the range of 223 to 230 nm. *Xylopia parviflora* fruits showed the presence of two major hydrophilic substances generating peaks at low RT values (1.5 min and 2.3 min) with UV absorption profiles typical of chlorogenic ester derivatives (λ_max_ = 325 nm), followed by a peak series with a Gaussian-like distribution shape and λ_max_ = 270 nm typical of gallotannin derivatives. The compositional profile of *Scorodophloeus zenkeri* was dominated by a major peak at RT = 10.4 min and UV absorption at λ_max_ = 273 nm, and a shoulder at λ = 320 nm with minor peaks at higher RT with similar spectroscopic profiles. *Monodora myristica* seeds were characterized by a major peak at RT = 12.2 min and λ_max1_ = 264 nm and λ_max2_ = 300 nm, typical of coumarin derivatives. The composition of *Tetrapleura tetraptera* fruits was characterized by a major peak at RT = 8.8 min with a spectroscopic profile like that of caffeic acid, but with shorter λ_max_ (312 nm), typical of amide caffeic derivatives. *Echinops giganteus* roots showed a major peak at RT = 11.4 min (λ_max_ = 268 nm and secondary maxima in the range 288-348 nm) and at least two additional minor peaks at lower RT (9.8 min and 7.8 min) with UV absorptions (λ_max_ = 246 nm, 321 nm, 335 nm), suggestive of substituted quinoline alkaloids. Gingerol, shogaol, and paradol (λ_max_ = 280 nm) were clearly detectable in the chromatographic profile of *Aframomum melegueta* fruits, while in that of *Dichrostachys glomerata*, several poorly resolved peaks were observed (λ_max_ = 220 nm, 276-280 nm). *Aframomum citratum* showed the presence of several components with different UV absorption profiles, among which that eluting at around RT = 7.1 min, with λ_max1_ = 210 nm and λ_max2_ = 280 nm was dominant. By contrast, the HPLC-UV-DAD profile of the *Zanthoxylum leprieurii* did not show any detectable peak (data not shown).

### 2.7. Effect of Spice Extracts on Oxidative-Stress-Modulated Glucose Uptake in HepG2 cells

Assessing uptake of glucose (2-deoxy-2-[(7-nitro-2,1,3-benzoxadiazol-4-yl) amino]-D-glucose (2-NBDG)) by cells is an established approach to evaluate insulin sensitivity and its modulation by exogenous agents. We initially explored the ability of spice extracts to modulate 2-NBDG uptake by HepG2 cells under basal conditions (Table 3). Insulin and metformin (both at 10 µM) significantly increased 2-NBDG uptake (+48% and +72%, respectively; *p* < 0.05 vs. control). Interestingly, a similar and significant (*p* < 0.05 vs. control) increase of glucose uptake was observed with four of the spice extracts (*Aframomum melegueta, Scorodophloeus zenkeri*, *Monodora myristica*, *Tetrapleura tetraptera*). Further, a concentration-dependent effect (range 1–20 µg/mL) was observed for the latter three spice extracts (Figure 6). In HepG2 cells, H_2_O_2_-induced oxidative stress resulted in a significant decrease of 2-NBDG uptake (−22% vs. control, *p* < 0.05), which was restored by the antioxidant agent Trolox (Table 4). Pretreatment of HepG2 cells with spice extracts resulted in a rescue of H_2_O_2_-induced decrease of glucose uptake, with a complete glucose uptake restoration by *Xylopia aethiopica*, *Xylopia parviflora*, *Scorodophloeus zenkeri*, *Monodora myristica*, and *Dichrostachys glomerata* (Table 4).

All plant extracts were used at 10 µg/mL and metformin at 10 µM = 1.3 µg/mL.

## 3. Discussion

Exploitation of novel nutraceuticals for the prevention and treatment of chronic cardiometabolic diseases like T2DM and the metabolic syndrome is currently very relevant and is the focus of intense research [15,16,17,18,19,20]. An important source of these products is represented by extracts derived from plants traditionally used in some African countries for nutritional and medicinal purposes [21], and currently under characterization in terms of biochemical and nutraceutical properties [6,8,9,10]. Moreover, since these plant extracts may exhibit a relevant antioxidant activity [6,9,10], they could represent a specific countermeasure to control excessive ROS generation, which is linked to insulin resistance in different pathological conditions (obesity, T2DM, and metabolic syndrome) [22,23].

Based on these observations, the present study aimed to evaluate the ability of selected traditional Cameroonian spices to counteract oxidative-stress-induced reduction of glucose uptake, using human HepG2 hepatoma cells as the experimental model. In addition, we implemented a comprehensive characterization of the antioxidant properties of these spice extracts. Among the 11 spice extracts included in the study, 8 were found to reduce H_2_O_2_-induced ROS production in HepG2 cells by 34–65%, and the most effective were *Xylopia aethiopica*, *Aframomum citratum, Echinops giganteus*, *Dichrostachys glomerata*, and *Xylopia parviflora*. The potential involvement of Nrf2, a master regulator of the transcription of genes involved in antioxidation, antioxidant biosynthesis, and metabolic shift, was then highlighted [24]. HepG2 cells showed a relatively high basal level of nuclear Nrf2, in agreement with previous studies [25], which was further increased by H_2_O_2_ exposure, but was not affected when the latter was preceded by treatment with either *Dichrostachys glomerata* or *Xylopia parviflora*. Interestingly, exposure to *Dichrostachys glomerata* alone resulted in a reduced Nrf2 accumulation, whereas *Xylopia parviflora* extract increased Nrf2 translocation, suggesting the involvement of different pathways of antioxidant homeostasis. The antioxidant capability of these spice extracts assessed by ORAC and FRAP analyses was in agreement with total phenol content, indicating the highest antioxidant power for the same extracts (in decreasing order: *Xylopia parviflora*, *Dichrostachys glomerata*, *Aframomum melegueta*, *Aframomum citratum*, *Echinops giganteus*, and *Tetrapleura tetraptera*). These results suggest that the antioxidant effects of phenolic compounds, and particularly polyphenols, present in some extracts could be the consequence of metal ion chelation and free radical scavenging [10,26]. In agreement with these findings, some studies have observed that extracts with higher phenolic compound levels also showed a higher antioxidant activity [27]. A similar trend was observed by Abdou et al. [28], Etoundi et al. [29], and Ene-Obong et al. [30], who studied the antioxidant effect of different spices. Similar experimental procedures and findings were also reported by Somanah et al. [31] when investigating the antioxidant effects of *Carica papaya* (Sol Var) on H_2_O_2_-induced oxidative stress in HepG2 cells. The antioxidant properties of phenolic compounds are directly related to their structure, which is most often composed of one (or more) aromatic rings carrying one or more hydroxyl groups, making them potentially capable of scavenging free radicals by forming resonance-stabilized phenoxyl radicals [32,33].

The phytochemical profiling of extracts deriving from the different species described indicated a wide compositional variability, with specimens (fruits or seeds) from some species showing the presence of a few or single dominating components (e.g., *Monodora myristica* seeds and *Tetrapleura tetraptera* fruits) and others containing a complex mix of structurally related and unrelated substances (e.g., fruits from *Xylopia parviflora* and *Aframomum citratum*). A few previous studies have reported inconsistent information regarding some of the plant parts from the species investigated in this study, and the details have been discussed elsewhere [12]. However, it is worth noting that the hydroalcoholic extract of *Xylopia parviflora*, which exhibited the simultaneous presence of potent antioxidant compounds such as caffeic acid derivatives and gallotannins, was the only spice extract found to be mostly active in all the assays employed in this study. These findings further support that its positive effects on the tested cell system are mediated by an antioxidant action against ROS. The results obtained suggest that the extracts act at the intracellular level and prevent the formation of free radicals, or that they can indirectly modulate the activity and expression of antioxidant and detoxifying enzymes [34,35]. In this study, we used HPLC-UV-DAD analysis to obtain fast and detailed information on the qualitative and quantitative differences existing among phytochemical extracts. However, a limitation of this technological approach is that it is not fully able to obtain immediate identification of all the observed species (especially when authentic standards are not available), which deserves further investigation.

Oxidative stress, as mentioned above, is correlated with insulin resistance on an individual basis [22,36], as also observed in an animal model with genetic deficiency of the H_2_O_2_-scavenging enzymes, which triggers adipose tissue oxidative stress and subsequent insulin resistance [37]. Moreover, several studies have demonstrated the contribution of increased ROS production to the development of diabetic vascular complications such as atherosclerosis [38]. Therefore, counteracting this pathophysiological event on a long-term basis may be beneficial for subjects at risk of T2DM as well as for patients with insulin resistance, T2DM, and metabolic syndrome, together with drug therapy if appropriate. Interestingly, the antioxidant properties of some Cameroonian plants, including 7 out of 11 spices considered here, have been evaluated in experimental models [12,27,28,29,30], although no information in this respect was available for the remaining four plants. Based on these considerations, the effect of spice extracts on oxidative-stress-induced reduction of glucose uptake, as an experimental paradigm of insulin resistance, was assessed in HepG2 cells. Four spice extracts (*Aframomum melegueta, Scorodophloeus zenkeri*, *Monodora myristica*, and *Tetrapleura tetraptera*) were found to increase glucose uptake, similarly to metformin at the concentration of 1.3 µg/mL. Moreover, pretreatment of HepG2 cells with extracts and subsequent H_2_O_2_ challenge unveiled the ability of *Xylopia aethiopica*, *Xylopia parviflora*, *Scorodophloeus zenkeri*, *Monodora myristica*, and *Dichrostachys glomerata* to fully rescue H_2_O_2_-induced reduction of glucose uptake, suggesting a potentially beneficial effect of these spices on insulin resistance through direct scavenging of ROS. The other spice extracts, although not fully effective, displayed some partial effectiveness in restoring normal glucose uptake.

The novelty of our findings is thus two-fold: (1) stimulation of glucose uptake is an important cellular event promoted by all spices tested, and (2) four spices (*Xylopia parviflora, Scorodophloeus zenkeri, Echinops giganteus*, and *Zanthoxylum leprieurii)* were shown to possess relevant antioxidant activity, in addition to that reported previously and herein for the remaining seven plants. These processes ultimately appear to be significant in terms of counteracting the impairment of glucose uptake produced by oxidative stress, thereby restoring insulin sensitivity. In support of our results, the hypoglycemic activity of four of the spices studied here, *Dichrostachys glomerata*, *Tetrapleura tetraptera, Xylopia aethiopica*, and *Aframomum melegueta*, was reported by other studies conducted in animal models of diabetes and in clinical trials [39,40,41,42,43].

Similarities and discrepancies among the findings of the present study and those from previous investigations on the same spice products may arise from different experimental protocols, especially different extraction procedures. The qualitative and quantitative analysis of biological molecules as well as the standardization of herbal medicines is a fundamental process to evaluate and possibly improve their safety, stability, and efficacy.

In conclusion, in the present study, we found that some Cameroonian spice extracts were able to rescue the ROS-induced intracellular impairment of glucose uptake. The observed potent antioxidant capability of most spice extracts may also be beneficial for several pathological conditions involving increased oxidative stress (chronic inflammation, atherosclerosis, cancer), in addition to improving insulin sensitivity, and may be coupled with our recent work showing their positive modulation of enzymes relevant to carbohydrate/lipid digestion and cardiometabolic diseases [12]. Moreover, the analytical data allowed the identification of some bioactive compounds that could justify the observed biological activities and that should be further characterized using more detailed analytical technologies. These results raise important questions for further research on spice extracts, aiming to validate their efficacy, toxicity, and safety with preclinical models and well-designed clinical trials, to demonstrate the health value of this nutraceutical approach, which may be placed beyond diet and before drug treatment, thus being complementary to pharmacological interventions.

## 4. Materials and Methods

### 4.1. Chemicals

Bovine insulin, (±)-6-hydroxy 2,5,7,8 tetramethylchromane-2-carboxylic acid (Trolox), oleic acid, 2′-azobis(2-methylpropionamidine) dihydrochloride (AAPH), Folin–Ciocalteu solution, Na_2_CO_3_, dimethylsulfoxide (DMSO), H_2_O_2_, gallic acid, and 2,2,6 tripyridyl-5-triazine (TPTZ) were acquired from Sigma, Saint-Louis, MO, USA. 2-Deoxy-2-[(7-nitro-2,1,3-benzoxadiazol-4-yl) amino]-D-glucose (2-NBDG) was acquired from Abcam, Cambridge, MA, USA; CM-H2DCFDA was acquired from Thermo Fisher Scientific, Rodano (MI), Italy. All reagents were of the highest grade available.

### 4.2. Spice Extracts Preparation

Plants material included 11 spices (*Xylopia aethiopica* (Dunal) A.Rich., *Xylopia parviflora* (A. Rich) Benth, *Scorodophloeus zenkeri* Harms, *Monodora myristica* (Gaertn.) Dunal, *Tetrapleura tetraptera* (Schum. & Thonn.) Taub., *Echinops giganteus* A.Rich., *Dichrostachys glomerata* (Forssk.) Chiov., *Afrostyrax lepidophyllus* Mildbr., *Aframomum melegueta* K.Schum., *Aframomum citratum* (C.Pereira) K.Schum., and *Zanthoxylum leprieurii* Guill. & Perr.)) harvested from various sites of the region of West Cameroon in September 2017. Samples included different fruits, seeds, and roots identified in the National Herbarium of Cameroon (http://irad.cm/national-herbarium-of-cameroun/) in Yaoundé (Cameroon) by comparison to the deposited specimens. Air-dried and powdered samples (100 g) were subjected to magnetic stirring with 100 mL of 70% hydroethanolic mixture for 4 and 16 h, respectively, at room temperature and in dark conditions. The mixture was filtered through Whatman™ cellulose filter paper, and the filtrate was concentrated under reduced pressure, then frozen with dry ice and alcohol to give residues which constituted the crude extracts. They were then placed at −20 °C and, after 5 min, lyophilized and stored at −20 °C for subsequent studies. Spice extract stock solutions (100 mg/mL) dissolved in DMSO were prepared, aliquoted, and kept at −80 °C. For experiments with HepG2 cells, extracts were dissolved in DMSO and added to the cells, giving a final concentration of DMSO not exceeding 0.1%.

### 4.3. Cell Cultures

The human hepatocellular carcinoma (HepG2, ATCC^®^ HB-8065 ™, Manassas, VA, USA) cell line was obtained from the American Type Culture Collection (ATCC^®^, Manassas, VA, USA) and grown as recommended. Cells were cultured in MEM (Eagle’s minimum essential medium) containing 2 mM L-glutamine, 100 μM non-essential amino acids, and 1 mM sodium pyruvate. 10% fetal bovine serum (FBS), 1% penicillin (100 U/mL), and streptomycin (100 μg/mL) were added. HepG2 cells were kept in culture in 100 mm diameter Petri dishes at 37 °C in an incubator set at 95% air and 5% CO_2_.

### 4.4. MTS Cell Viability Assay

Cell viability assay was measured using the Cell Titer 96 aqueous non-radioactive cell proliferation assay (Promega, Madison, WI, USA) according to the method described by Kim and Jang [44] with slight modifications. In brief, HepG2 cells were seeded in a sterile flat-bottomed 96 well plate and incubated at 37 °C for 24 h in a humidified incubator containing 5% CO_2_. Different concentrations of extracts (1, 10, 50, and 100 µg/mL) were prepared directly in fresh serum-free MEM media and 100 µL of each treatment was added to each well and incubated for 24 and 48 h. Next, 20 µL of the MTS reagent in combination with the electron coupling agent phenazine methosulfate was added into each well and allowed to react for 1 h at 37 °C. After 2 min of shaking at minimal intensity, the absorbance at 490 nm was measured using the EnSpire PerkinElmer Multimode Plate Reader. Controls (cells with media containing DMSO (≤ 0.1%)) and blanks (wells containing media without cells) were assessed under the same conditions. The cell viability values were determined using the equations below.
% cell viability = ((mean sample absorbance - mean blank absorbance)/(mean control absorbance - mean blank absorbance)) × 100(1)

Three separate experiments were conducted, each consisting of three distinct readings.

### 4.5. Morphological Analysis

Cells were cultured as reported above in sterile flat-bottomed 6 cm^2^ dishes and incubated at 37 °C for 24 h in a humidified incubator containing 5% CO_2_. Two different concentrations of extracts (1 and 100 µg/mL) were prepared directly in fresh serum-free media and 3 mL of each treatment was added to each dish and incubated for 24 h and 48 h. After the treatment, cells were visualized on a ZEISS microscope (ZEISS, VA, USA) using 10× and 32× magnifications.

### 4.6. ROS Modulation Analysis in Cells

A fluorometric assay was used to determine the intracellular ROS level in HepG2 cells, using the oxidant-sensitive fluorescence probe CM-H2DCFDA according to the method described by Somanah et al. [31]. Cells were seeded in 96 well black plates, and, after reaching 90% confluence, incubated with spice extracts at a different concentrations (0–20 µg/mL) for 24 h. Cells were then exposed to 20 µM CM-H2DCFDA and subsequently incubated with 500 µM hydrogen peroxide for 1 h to induce ROS production. The resulting fluorescence intensities were measured at an excitation wavelength of 485 nm and an emission wavelength of 535 nm using the EnSpire PerkinElmer Multimode Plate Reader [31].

### 4.7. Determination of the Total Phenolic Content of the Extracts

The determination of the phenols was carried according to the method previously described by Anim et al. [45], with minor modifications. An aliquot of 50 μL of extract stock solution (1.0 mg/mL) previously diluted in 750 μL of distilled water was introduced into a test tube. Next, 50 μL of 2N Folin–Ciocalteu solution was added, followed by incubation at room temperature for 4 min. Freshly prepared Na_2_CO_3_ (20%) solution (150 μL) was added and the tubes were further incubated for 30 min at 40 °C. Absorbance was measured at 755 nm and for each sample, three tests and three independent analyses were performed in the same conditions. The concentration calibration range from 0 to 30 μg/mL was performed using a 1.0 mM gallic acid stock solution following sample procedure. The results were expressed in μg of gallic acid equivalent/g of extract and in percentage of phenolic compounds expressed as gallic acid equivalent. The final concentration of the extracts was 50 μg/mL.

### 4.8. Oxygen Radical Absorbance Capacity Assay

The oxygen radical absorbance capacity (ORAC) test was performed according to the method previously described in Reference [46] with minor modifications. An aliquot of 20 μL of 1 µg/mL stock solution of each extract was introduced into a polypropylene fluorescence 96 well microplate. Subsequently, 120 µL of fluorescein solution (70 nM final concentration) previously prepared with a phosphate buffer (pH 7.4) was added to each well. The final concentration of each extract was 0.1 μg/mL. The microplate was incubated for 15 min at (37 °C and 60 μL of an AAPH reagent solution (40 mM) previously prepared with phosphate buffer (pH 7.4) was rapidly added. After 10 s of shaking at maximum intensity, the microplate was immediately placed in a microplate reader and the fluorescence was recorded every 2 min for 60 min at 485 nm (length excitation wave) and at 528 nm (emission wavelength). Each sample was analyzed three times in the same conditions and three distinct readings were performed. The calibration curve (concentrations ranging from 0–50 µM using 0.004 M Trolox stock solution) was then prepared. The calibration curve was prepared using 0.004 M Trolox stock solution following sample procedure. The ORAC values were calculated using the area under the curve (AUC) and net AUC of the standards and samples, determined using the equations below. Results are expressed in μM Trolox equivalent/g of extract.
AUC = 0.5+(R2/R1)+(R3/R1)+(R4/R1)+……+ 0.5(Rn/R1)(2)
where R1 is the fluorescence reading at the initiation of the reaction and Rn is the last measurement.
Net AUC = AUC standard/sample − AUC blank(3)

### 4.9. Ferric-Reducing Antioxidant Power Assay

The ferric-reducing antioxidant power (FRAP) assay, adapted from Reference [47], was modified to evaluate the reducing power of spice extracts. At low pH, ferric tripyridyltriazine complex is reduced to its ferrous form, the resulting intense blue color being linearly related to the amount of reductant present. The FRAP reagent consisting of 2,2,6 tripyridyl-5-triazine (TPTZ, 10 mM) in 40 mM HCl and 20 mM ferric chloride in 200 mL of sodium acetate buffer (pH 3.6, 0.25 M) was freshly prepared and warmed at 37 °C prior to analysis. Two milliliters of FRAP reagent and 75 μL of the extracts were left to react for 20 min at room temperature (final concentration of the extracts: 37 μg/mL). The absorbance was read at 593 nm. The concentration calibration range from 0 to 250 μg/mL was prepared using a 1.0 mM gallic acid stock solution following sample procedure. The results were expressed as μg of gallic acid equivalent/g of extract.

### 4.10. Nrf2 Immunofluorescence Assay

Nrf2 localization was assessed by immunofluorescence (IF) analysis. Cells were grown on 12 mm polylysine-coated glass slides and then treated with 10 µg/mL spice extract and/or 500 µM H_2_O_2_ (without FBS). After 4 h at 37 °C in a humidified incubator containing 5% CO_2_, cells were fixed in 4% formaldehyde for 5 min at RT and the subcellular localization of the Nrf2 was examined by IF staining. The formaldehyde-fixed cells were preincubated for 20 min with 10% normal serum in PBS and then incubated with a diluted solution of the primary antibody (1:100; Invitrogen, cat. PA527882) for 1 h at RT. Cells were then washed in PBS and incubated for 1 h with the Alexa Fluor 488 F(ab’)2 fragment of anti-rabbit antibody (1:400; Invitrogen, cat. A32731). Nuclei were counterstained with 4′,6-diamidino-2-phenylindole dihydrochloride (DAPI) (Fluoroshield Sigma-Aldrich, St. Louis, MO, USA). For negative controls, the primary antibody was omitted. Preparations were then examined with a fluorescent microscope (magnification 32×; Carl Zeiss, Thornwood, NY, USA) and images were recorded.

### 4.11. HPLC-UV-DAD Analysis

Separation of the compounds present in extracts was carried out using a Varian LC-940 high-performance liquid chromatography (HPLC) instrument (Varian, Turin, Italy) coupled to a diode array detector (DAD). The extracts were solubilized in a hydroalcoholic solvent system (20 mg/mL; 70% w/vol) and then sonicated for sixty minutes (60 min). After stirring, the mixture was centrifuged at 4000 rpm for 10 min and filtered through paper (Whatman™ paper No 1) to separate insoluble material. The obtained solutions were separated on a Kinetex™ biphenyl 100 Å (100 × 4.6 mm; 2.6 μm) column and the elution gradient was performed with 0.05% aqueous formic acid as mobile phase A and 0.05% formic acid in acetonitrile as phase mobile B, at a flow rate of 1.6 mL/min. The elution gradient used for the separation was 5% B 0–3 min; from 5% B to 60% B 3–15 min; from 60% B to 80% 15–20 min. The sample tray temperature and the injection volume were, respectively, set at 4 °C and 5 μL. The acquisition wavelength was adjusted depending on the specific sample.

### 4.12. Glucose Uptake Assay

Glucose uptake activity was analyzed by measuring the uptake of 2-NBDG. Cells were cultured in 96 well black plates for 48 h. Cells were then incubated with spice extracts (1–20 µg/mL) or 500 µM metformin for 24 h. Finally, 500 µM H_2_O_2_ was then added for the last incubation hour. Subsequently, to determine 2-NBDG uptake, cells were incubated with 80 µM 2-NBDG and 0.1 mM insulin (dissolved in glucose-free medium) for 1 h, and then were washed twice with ice-cold PBS to stop further uptake. 2-NBDG fluorescence intensities were measured using a microplate reader (EnSpire, PerkinElmer) at excitation and emission wavelengths of 465 and 540 nm, respectively. Glucose uptake increase rate was calculated as follows.
Glucose uptake increase rate (%) = ((FI sample - FI Blank)/(FI Control - FI Blank)) × 100(4)

FI: fluorescent signal.

### 4.13. Statistical Analysis

Results from at least three independent experiments carried out in triplicate or quadruplicate were expressed as mean (± SD) values or as a mean percentage (%) compared to a control. Mean differences were determined by one-way ANOVA using SPSS (Version 25). The Waller–Duncan test was used to test differences of means. Pearson’s linear test was used to test for correlation and values of less than 0.05 were considered statistically significant. All graphs were generated using GraphPad Prism (Version 7).

## Figures and Tables

**Figure 1 metabolites-10-00182-f001:**
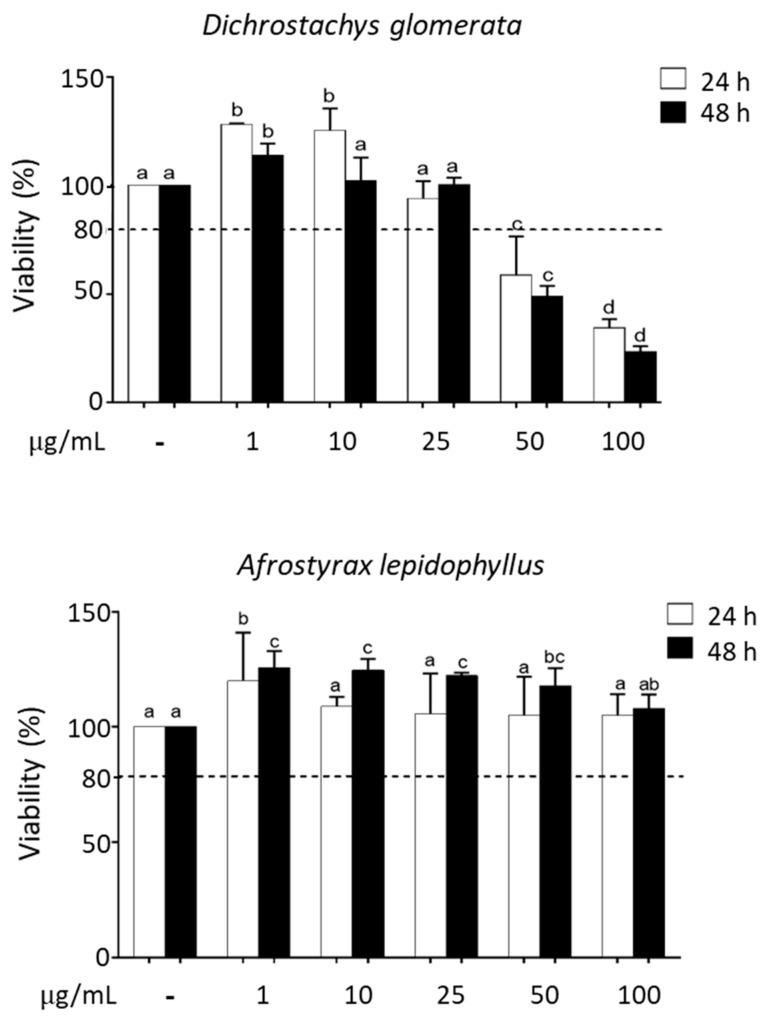
Cytotoxicity of spice extracts in HepG2 cells evaluated via [3-(4,5-dimethylthiazol-2-yl)-5-(3-carboxymethoxyphenyl)-2-(4-sulfophenyl)-2H-tetrazolium] (MTS) assay. Effects of different concentrations (1–100 µg/mL) of *Dichrostachys glomerata* and *Afrostyrax lepidophyllus* extracts on cell viability. Incubation time: 24 and 48 h. Data are expressed as % of control taken as 100; mean ± SD, N = 3. The dotted line set at 80% indicates the threshold above which cells are considered viable. Different letters (a, b, c, d) refer to significant differences among values at the 5% probability threshold (Waller–Duncan test). Viability data for all spice extracts are shown in Appendix A.

**Figure 2 metabolites-10-00182-f002:**
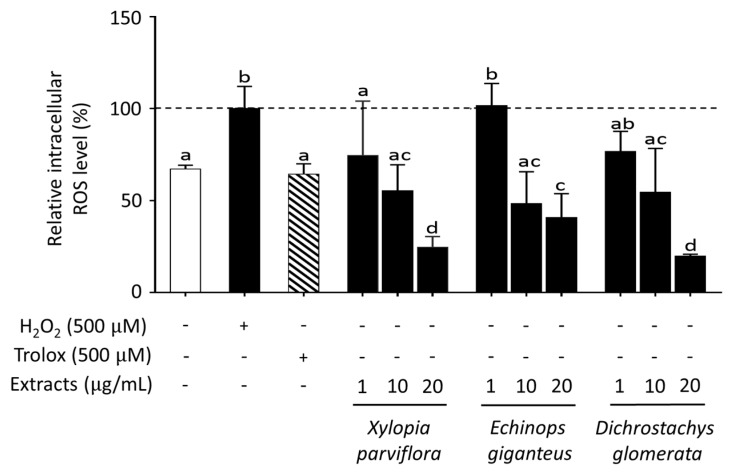
Effect of spice extracts on intracellular reactive oxygen species (ROS) production in HepG2 cells. Effects of different concentrations (1–20 µg/mL) of *Xylopia parviflora, Echinops giganteus*, and *Dichrostachys glomerata* extracts on intracellular ROS production. Incubation time: 24 h of pretreatment and treatment with H_2_O_2_ for the last 1 h. Data are expressed as % of control taken as 100; mean ± SD, N = 3. Different letters (a, b, c, d) refer to significant differences among values at the 5% probability threshold (Waller–Duncan test).

**Figure 3 metabolites-10-00182-f003:**
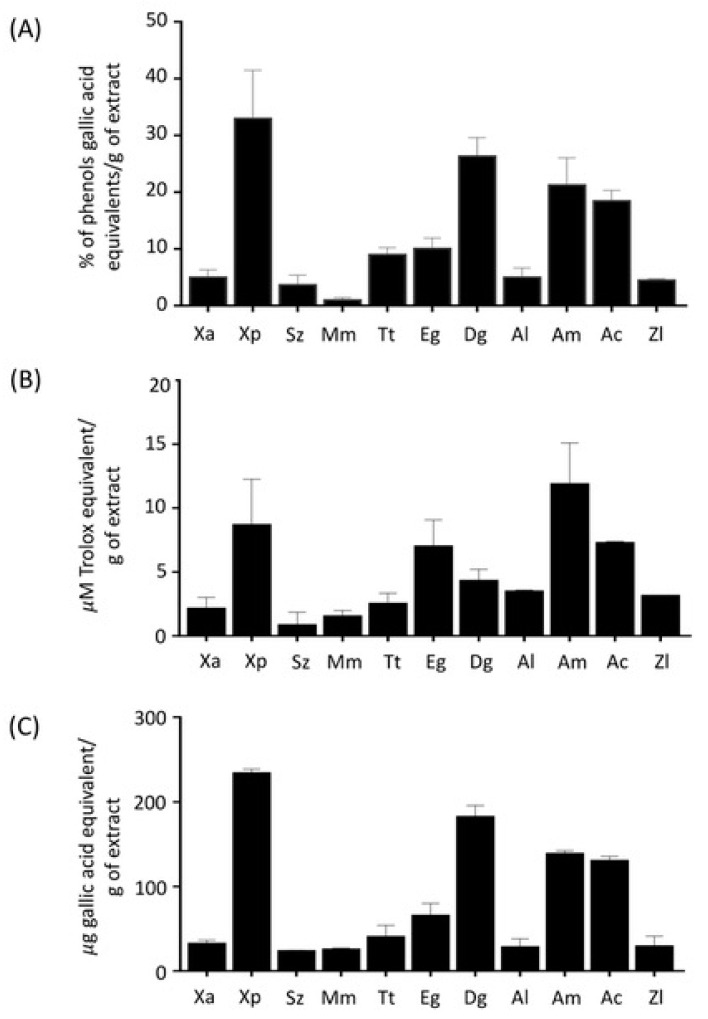
Antioxidant properties of spice extracts in HepG2 cells. Antioxidant capacity of the extracts was evaluated by (**A**) phenol content quantification (data are expressed as % of phenols gallic acid equivalents/g of extract), (**B**) oxygen radical absorbance capacity (ORAC) (data are expressed as μM Trolox equivalent/g extract), and (**C**) ferric-reducing antioxidant power (FRAP) assays (data are expressed as μg of gallic acid equivalent/g of extract). Different letters (a, b, c, d) refer to significant differences among values at the 5% probability threshold (Waller–Duncan test). Abbreviations: *Xylopia aethiopica* (Xa), *Xylopia parviflora* (Xp), *Scorodophloeus zenkeri* (Sz), *Monodora myristica* (Mm), *Tetrapleura tetraptera* (Tt), *Echinops giganteus* (Eg), *Afrostyrax lepidophyllus* (Al), *Dichrostachys glomerata* (Dg), *Aframomum melegueta* (Am), *Aframomum citratum* (Ac) *and Zanthoxylum leprieurii* (Zl).

**Figure 4 metabolites-10-00182-f004:**
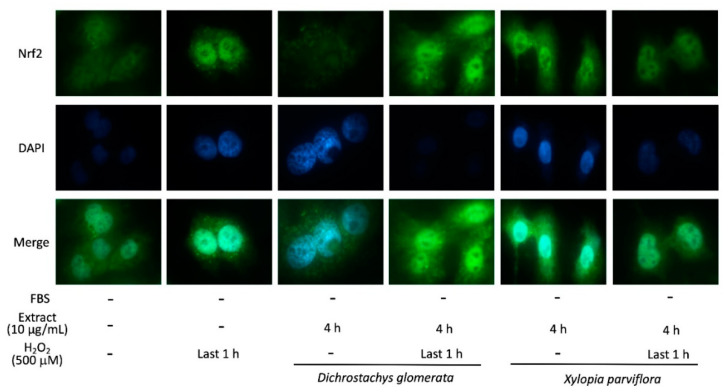
Effect of *Dichrostachys glomerata* and *Xylopia parviflora* extracts on Nrf2 nuclear translocation in HepG2 cells. Cells were treated with 10 µg/mL spice extracts for 4 h; 500 µM H_2_O_2_ was added in the last hour. Nrf2 was detected with Alexa Fluor 488 F(ab’)2 fragment of anti-rabbit antibody; nuclei were counterstained with 4′,6-diamidino-2-phenylindole dihydrochloride (DAPI). Images are representative of three separate observations.

**Figure 5 metabolites-10-00182-f005:**
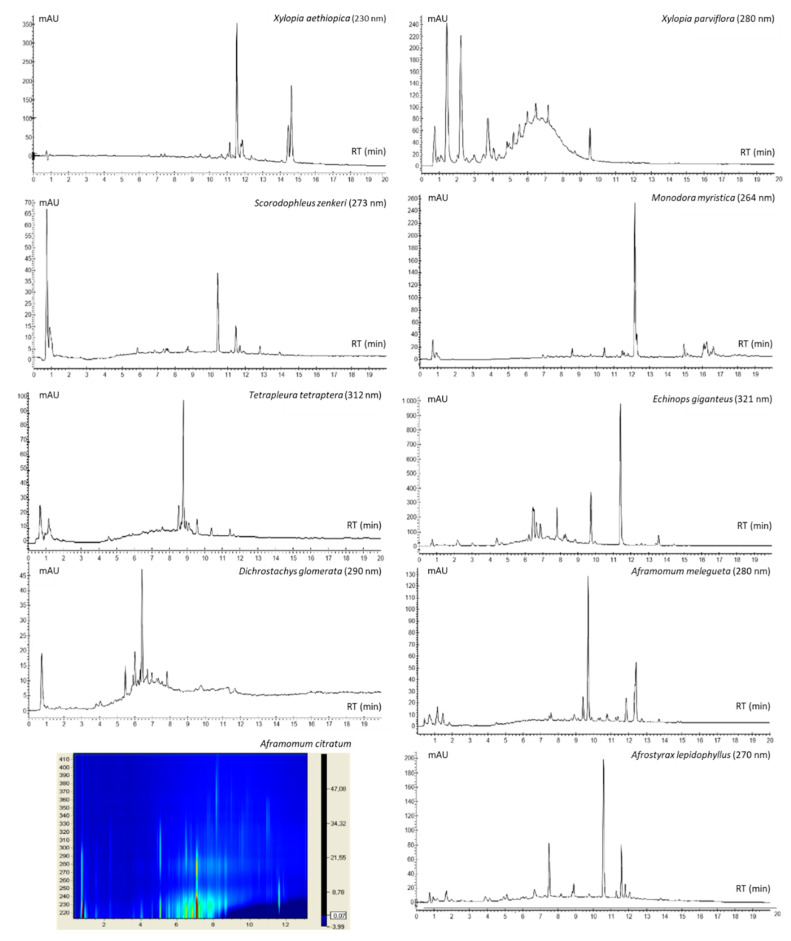
Reverse-phase high-performance liquid chromatography coupled to a diode array detector (HPLC-UV-DAD) analysis of spice extracts. Chromatogram of the components of *Xylopia aethiopica*, *Xylopia parviflora*, *Scorodophloeus zenkeri*, *Monodora myristica*, *Tetrapleura tetraptera*, *Echinops giganteus*, *Dichrostachys glomerata*, *Aframomum melegueta*, *Afromomum citratum*, and *Afrostyrax lepidophyllusii* extracts.

**Figure 6 metabolites-10-00182-f006:**
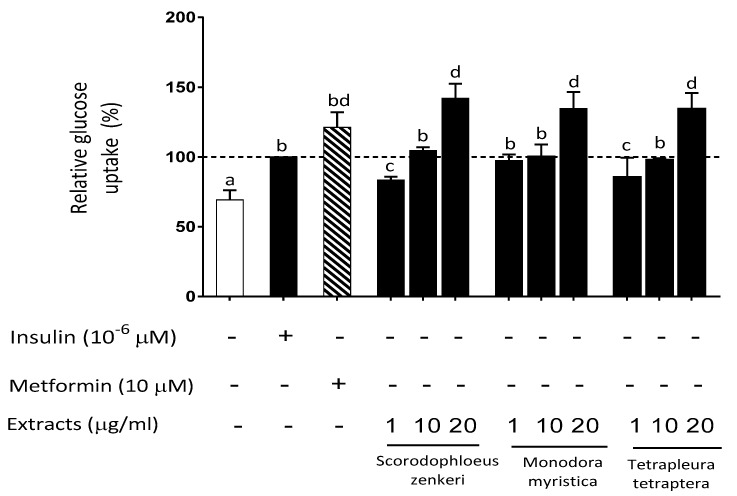
Effect of spice extracts on glucose uptake. Concentration-dependent (1–20 µg/mL) stimulation of glucose uptake in HepG2 cells with *Scorodophloeus zenkeri*, *Monodora myristica*, *Tetrapleura tetraptera*, and metformin (10 µM = 1.3 µg/mL). Incubation time: 24 h. Data are expressed as % of insulin treatment taken as 100; mean ± SD, N = 4. Different letters (a, b, c, d) refer to significant differences among values at the 5% probability threshold (Waller–Duncan test).

**Table 1 metabolites-10-00182-t001:** Identification and yield of plant extraction.

Names of Plants	Family	Herbarium Voucher Number	Part Used	Extract Aspect	Extract Color	Extraction Yield (%)
*Xylopia aethiopica* (Dunal) A. Rich	Annonaceae	16419/SRF-Cam**	Fruits	Powder	Brown-strand	23.9
*Xylopia parviflora* (A. Rich.) Benth	Annonaceae	6431/ SRF-Cam	Seeds	Powder	Brown-beige	20.5
*Scorodophloeus zenkeri* Harms	Fabaceae	44803/HNC*	Seeds	Crystal	Brown-auburn	16
*Monodora myristica* (Graertm.) Dunal	Annonaceae	2949/ SRF-Cam	Seeds	Oil	Yellow-saffron	27.9
*Tetrapleura tetraptera* (Schum. & Thonn.)Taub	Fabaceae	12117/SR-Cam	Fruits	Powder	Brown-bistra	49.2
*Echinops giganteus* Var Lellyi C. D. Adams	Asteraceae	23647/SRF-Cam	Roots	Powder	Yellow-topaz	13.1
*Afrostyrax lepidophyllus* Mildbr	Huaceae	44853/HNC	Seeds	Crystal	Yellow-amber	7.1
*Dichrostachys glomerata* (Forssk.) Hutch	Fabaceae	15220/SRF-Cam	Seeds	Crystal	Brown-coffee	27.7
*Aframomum melegueta* (Roscoe) K.Schum	Zingiberaceae	39065/HNC	Fruits	Powder	Brown-acajou	11.5
*Aframomum citratum* (Pereira ex Oliv. and Hanb) K. Shum.	Zingiberaceae	37736/HNC	Fruits	Powder	Beige	6.4
*Zanthoxylum leprieurii* Guill. Et Perr.	Rutaceae	37632/HNC	Seeds	Powder	Brown-bistra	32.7

*HNC: Cameroun National Herbarium; **SRF-Cam: Société de Réserves Forestière du Cameroun.

**Table 2 metabolites-10-00182-t002:** Relative intracellular ROS production in HepG2 cells: effect of spice extracts.

	H_2_O_2_ (500 µM)	Intracellular ROS Level (% of H_2_O_2_-treated cells)
**Control**	-	49.1±18.2^a^
**H_2_O_2_ (500 µM)**	+	100^b^
**Trolox (500 µM)**	+	69.0±15.0^c^
***Xylopia aethiopica***	*+*	35.4±14.0^d^
***Xylopia parviflora***	*+*	37.8±8.3^d^
***Scorodophloeus zenkeri***	*+*	73.2±15.0^ab^
***Monodora myristica***	*+*	82.6±11.4^ab^
***Tetrapleura tetraptera***	*+*	66.2±17.5^c^
***Echinops giganteus***	*+*	50.9±16.3^ac^
***Afrostyrax lepidophyllus***	*+*	105.8±17.6^b^
***Dichrostachys glomerata***	*+*	43.5±8.3^a^
***Aframomum melegueta***	*+*	56.6±9.0^ac^
***Aframomum citratum***	*+*	37.2±15.4^d^
***Zanthoxylum leprieurii***	*+*	52.3±19.5^c^

Data are expressed as % of control taken as 100; mean ± SD, N = 3. Different letters (a, b, c, d) refer to significant differences among values at the 5% probability threshold (Waller–Duncan test). All spice extracts were used at 10 µg/mL.

**Table 3 metabolites-10-00182-t003:** Glucose uptake in HepG2 cells: effect of spice extracts.

	Glucose Uptake (% of Insulin)
**Control**	52.31±6.55^a^
**Insulin (10 µM)**	100^b^
**Metformin (10 µM)**	124.81±3.10^c^
***Xylopia aethiopica***	68.15±8.64^ad^
***Xylopia parviflora***	57.64±6.54^a^
***Scorodophloeus zenkeri***	104.59±1.39^cd^
***Monodora myristica***	100.55±7.62^cd^
***Tetrapleura tetraptera***	98.05±1.54^cd^
***Echinops giganteus***	69.03±4.54^ad^
***Afrostyrax lepidophyllus***	55.55±3.49^a^
***Dichrostachys glomerata***	60.29±5.76^a^
***Aframomum melegueta***	70.15±6.37^d^
***Aframomum citratum***	58.82±8.56^a^
***Zanthoxylum leprieurii***	56.84±7.11^a^

Data are expressed as % of insulin-treated cells taken as 100; mean ± SD, N = 3. Different letters (a, b, c, d) refer to significant differences among values at the 5% probability threshold (Waller–Duncan test).

**Table 4 metabolites-10-00182-t004:** Effect of plant extracts on oxidative stress modulation of glucose uptake in HepG2 cells.

	H_2_O_2_ (500 µM)	Glucose Uptake (% of Control)
**Control**	-	100±7.2^a^
**H_2_O_2_ (500 µM)**	+	77.7± 2.6^b^
**Trolox (500 µM)**	+	98.9±12.6^a^
***Xylopia aethiopica***	*+*	100.5±10.1^a^
***Xylopia parviflora***	*+*	102.6±25.0^a^
***Scorodophloeus zenkeri***	*+*	100.4±14.7^a^
***Monodora myristica***	*+*	100.5±2.8^a^
***Tetrapleura tetraptera***	*+*	90.0±16.9^a^
***Echinops giganteus***	*+*	89.3±20.3^a^
***Afrostyrax lepidophyllus***	*+*	91.0±20.3^a^
***Dichrostachys glomerata***	*+*	98.6±20.9^a^
***Aframomum melegueta***	*+*	95.1±16.1^a^
***Aframomum citratum***	*+*	87.1±14.7^a^
***Zanthoxylum leprieurii***	+	91.7±12.3^a^

Incubation time: 24 h; 500 µM H_2_O_2_ was added in the last hour. Data are expressed as % of control taken as 100; mean ± SD, N = 3. Different letters (a, b) refer to significant differences among values at the 5% probability threshold (Waller–Duncan test). All plant extracts were used at 10 µg/mL.

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
