# Peer review of "Oxidative Stress Modulation by Cameroonian Spice Extracts in HepG2 Cells: Involvement of Nrf2 and Improvement of Glucose Uptake"

_metabolites, 2020, doi:10.3390/metabo10050182_

Round 1

Reviewer 1 Report

The authors describe a series of experiments that examine the hypothesis that Cameroonian spice extracts can modulate oxidative stress responses in HepG2 cells.

This study should be of interest to researchers interested in modulation of cellular responses by plant metabolites. This is well written manuscript that explores biological activities of Cameroonian spices. It would have been appreciated if the analysis of the spice extracts had been by LC-MS which would have improved the specificity of the analysis but this can be overlooked if such an analysis is not available. Otherwise I enjoyed reading this paper and I found no typos.

Figure 2. In my copy the µ symbol was replaced with a square box.

Author Response

The authors describe a series of experiments that examine the hypothesis that Cameroonian spice extracts can modulate oxidative stress responses in HepG2 cells.

This study should be of interest to researchers interested in modulation of cellular responses by plant metabolites. This is well written manuscript that explores biological activities of Cameroonian spices.

Thank you for your positive evaluation.

It would have been appreciated if the analysis of the spice extracts had been by LC-MS which would have improved the specificity of the analysis but this can be overlooked if such an analysis is not available. Otherwise I enjoyed reading this paper and I found no typos.

Thanks. We understand that LC-MS may give further details and could be an additional study to be done in the future. However, complementary data about the extracts compositional fingerprinting by post-derivatisation GC-EIMS analysis has been published last year and cited in the manuscript as paper “Atchan Nwakiban et al. 2019”.

Figure 2. In my copy the µ symbol was replaced with a square box.

Unfortunately, this is a common formatting problem between the macOS system and Microsoft, in the PDF format the problem does not persist. We edited this image as well as 2 others with the same problem (Figures 1,2,4) to be sure that the units were visible for all the operating systems (line 110).

Reviewer 2 Report

In this report, Nwakiban et al., report on protective effects of Cameroonian-spice extracts and oxidative stress.  Some comments for improvement are provided below.  A large number of the spices tested as extracts showed anti-oxidant qualities as assessed in a hepatic cell line in response to the oxidant H2O2. Spices were also tested for toxicity in the cell line and most studies conducted at less than 25 ug/ml. Further analysis showed the polyphenol content of the spice(s) was associated with its anti-oxidant capabilities. Xylopia parviflora and Dichrostachys glomerata were the most promising on these initial measures. Nrf2 translocation to the nucleus was studied in response to these two spices, but yielded opposite results.   HPLC-UV-DAD was conducted to further characterize the spices, and resulting chromatograms described.  Four of the spices were able to increase glucose uptake into cells.  H2O2 treatment of cells reduced glucose uptake, which could be restored by this subclass of spices.  Overall, a well-written paper and a study revealing some potential means whereby these traditional herbs may be protective.  I have only minor suggestions.

  1. Results, Line 127, why was such a high concentration 1 mg/ml of spice extracts used for FRAP analysis.
  2. A rationale for the use of the hepatocarcinoma cell line should be provided.
  3. Page 71 (table)- in the row describing the herb, tetrafleura tetraptera, it is unclear what material is being extracted, “uits”?
  4. Methods- please define acronym “MTS” at its first usage.  
  5. Line 95-96- should be “to test this latter effect” (correct tense).
  6. To bolster findings, anti-oxidant properties of these spices could be tested in a second cell line.

Author Response

In this report, Nwakiban et al., report on protective effects of Cameroonian-spice extracts and oxidative stress.  Some comments for improvement are provided below.  A large number of the spices tested as extracts showed anti-oxidant qualities as assessed in a hepatic cell line in response to the oxidant H2O2. Spices were also tested for toxicity in the cell line and most studies conducted at less than 25 ug/ml. Further analysis showed the polyphenol content of the spice(s) was associated with its anti-oxidant capabilities. Xylopia parviflora and Dichrostachys glomerata were the most promising on these initial measures. Nrf2 translocation to the nucleus was studied in response to these two spices, but yielded opposite results.   HPLC-UV-DAD was conducted to further characterize the spices, and resulting chromatograms described.  Four of the spices were able to increase glucose uptake into cells.  H2O2 treatment of cells reduced glucose uptake, which could be restored by this subclass of spices.  Overall, a well-written paper and a study revealing some potential means whereby these traditional herbs may be protective.  I have only minor suggestions.

Thanks for your positive evaluation.

  1. Results, Line 127, why was such a high concentration 1 mg/ml of spice extracts used for FRAP analysis.

Thank you very much for this comment, highlighting an error in reporting tested concentration values for all 3 assays reported in Fig. 3.

Phenol content assay: the extracts were assayed at the final concentration of 50 µg/mL.

ORAC assay: the extracts were assayed at the final concentration of 0.1 µg/mL.

FRAP assay: the extracts were assayed at the final concentration of 37 µg/mL.

We have modified the text accordingly (lines 123, 129, 392-395, 398, 401, 405, 408, 409, 421-424).

Figure 3 (corresponding to line 136) has also been modified and the 3 words “Extracts” and the relative (erroneous) concentrations were removed.

  1. A rationale for the use of the hepatocarcinoma cell line should be provided.

Thanks for this comment. We used the human HepG2 cell line in this study for several reasons. This cell line is a widely used experimental model of human liver cell, retaining most biochemical features of normal hepatocytes. It has been utilized to assess liver toxicity of many agents, since it expresses a full set of CYP450, similarly to normal liver. According to the literature, it has been used for a relevant (more than 30) number of studies dealing with oxidative stress and glucose uptake/insulin resistance. Accordingly, at line 64-66, we added as follows: “To this purpose, we selected the human hepatoma HepG2 cell line, which is a widely used experimental model of human liver cell, often utilised in several studies on oxidative stress and glucose uptake/insulin resistance [11].”

  1. Page 71 (table)- in the row describing the herb, tetrafleura tetraptera, it is unclear what material is being extracted, “uits”?

Thanks. The correct word is “fruits”. Corrected in the revised version (line 72).

  1. Methods- please define acronym “MTS” at its first usage.  

Thanks, sorry. “MTS” is the acronym of [3-(4,5-dimethylthiazol-2-yl)-5-(3-carboxymethoxyphenyl)-2-(4-sulfophenyl)-2H-tetrazolium]. This has been added in the Results section when was first cited (line 74).

  1. Line 95-96- should be “to test this latter effect” (correct tense).

Thanks, we corrected the error (lines 96/97).

  1. To bolster findings, anti-oxidant properties of these spices could be tested in a second cell line.

Thanks, this is a very important point. Toward the end of this study with HepG2 cells, we actually started to evaluate the antioxidant properties of these spices in other cell lines relevant for metabolic diseases. When completed, these findings will be the focus of another manuscript.

Reviewer 3 Report

The authors describe the effect of several Cameroonian spice extracts on ROS levels and ROS-induced impairment of glucose uptake in HepG2 cells. There are several studies of some of these extracts in the literature a few featured extensively (e.g. Xylopia aethiopica, Aframomum melegueta, Monodora myristica) in relation to cellular protective effects. These published studies (some having gone to much great length in identifying bioactive compounds) detract from the novelty and originality of the presented work. There are additional issues with the methodology and conclusions of the studies presented that also diminish the significance of the work described here. Some of these issues are:

  1. Fig S1 and S2 not made available for review, not sure if this is an editorial issue or something else.
  2. Fig. 1 units X-axis cannot be g/ml perhaps microgram/mL. Same issue, units for treatment dose employed were not visible on Fig 2 and 4 due to a formatting problem.
  3. The studies illustrated in Fig. 4 are non conclusive. The source, catalogue # of the primary antibody is not listed, the number of observations is not indicated. Are these images representative of all observations? Even as they are, the images are of very poor quality. It is not clear if the antibody used actually detects NRF2, it would need to be validated by immunoblotting or in cells with overexpression or supressed expression of NRF2. Secondly, the effect of the extracts on the subcellular localization of Nrf2, should be verified via an orthogonal, complementary approach like subcellular fractionation or by examining the expression of known gene targets of Nrf2.
  4. Fig 5, the description of the HPLC-UV-DAD profiles of various plant extracts is not very useful or informative unless the authors individually interrogate the antioxidant effect of compounds corresponding with specific peaks or pooled fractions.
  5. It is not clear what is meant by "the HPLC-UV-DAD profile of the Zanthoxylum leprieurii did not show any detectable relevant peak". Not all compounds absorb light with a sufficiently high absorption coefficient to allow for UV-detection (eg saturated fatty acids). What is a peak supposed to be relevant to if it is not functionally implicated?
  6. Though the studies generally progress fairly logically, the link to the last set of experiments on the effect of extracts on glucose uptake seems a little tenuous. There is no reason why the effects on glucose uptake should be related to the effects of these extracts on oxidative stress alone.
  7. Unclear why the effect of extracts on oxidative stress-modulated glucose uptake in HepG2 cells has such high variability and how this variability with n of only 3 still allowed for the rejection of the null hypothesis.
  8. Instead of superficially characterizing a multitude of extracts from different plants, it would have been more beneficial to perform a systematic in-depth characterization of selected few, and perhaps make some attempt to isolate and chemically characterize the bioactive component(s) of these extracts.

minor: non-standard abbreviations need to be spelled out on first occurence (MTS

Author Response

The authors describe the effect of several Cameroonian spice extracts on ROS levels and ROS-induced impairment of glucose uptake in HepG2 cells. There are several studies of some of these extracts in the literature a few featured extensively (e.g. Xylopia aethiopica, Aframomum melegueta, Monodora myristica) in relation to cellular protective effects. These published studies (some having gone to much great length in identifying bioactive compounds) detract from the novelty and originality of the presented work.

There are additional issues with the methodology and conclusions of the studies presented that also diminish the significance of the work described here. Some of these issues are:

  1. Fig S1 and S2 not made available for review, not sure if this is an editorial issue or something else.

Thanks, Fig S1 and S2 were uploaded with the rest of the files. It is not clear to us why they were not made available for the review. In any case they have been uploaded also with the revised ms.

  1. 1 units X-axis cannot be g/ml perhaps microgram/mL. Same issue, units for treatment dose employed were not visible on Fig 2 and 4 due to a formatting problem.

Thanks for this comment. Unfortunately, this is a common formatting problem between the macOS system and Microsoft, in the PDF format the problem does not persist. We edited the images to be sure that the units were visible for all the operating systems (lines 87/110).

  1. The studies illustrated in Fig. 4 are non conclusive. The source, catalogue # of the primary antibody is not listed, the number of observations is not indicated. Are these images representative of all observations? Even as they are, the images are of very poor quality. It is not clear if the antibody used actually detects NRF2, it would need to be validated by immunoblotting or in cells with overexpression or supressed expression of NRF2. Secondly, the effect of the extracts on the subcellular localization of Nrf2, should be verified via an orthogonal, complementary approach like subcellular fractionation or by examining the expression of known gene targets of Nrf2.

Thanks for this important comment. The source and the catalogue # of the primary antibody have now been listed (lines 431/432). The number (=3) of observations has been reported in the legend to Fig. 4 and one representative observation/experiment is shown (line 153). In Fig. 4, the images have been now processed using a dedicated software (Affinity Photo 1.8.3) in order to homogeneously increase the quality (line 153).

We selected this specific antibody against Nrf2 since it has been successfully used in several studies (for example: Yang et al. Acetyl-l-carnitine prevents homocysteine-induced suppression of Nrf2/Keap1 mediated antioxidation in human lens epithelial cells., Molecular Medicine Reports 2015, 12 (1): 1145-50; Akhtar et al. Acute maternal oxidant exposure causes susceptibility of the fetal brain to inflammation and oxidative stress. Journal of Neuroinflammation 2017, 14 (1): 195).

We also planned to do experiments evaluating Nrf2 localization through subcellular fractionation followed by ELISA, but unfortunately these activities were not (and still are not) possible after the advent of Sars-Cov-2 pandemic and the subsequent lockdown of our laboratories. At the moment we cannot indicate any possible date for the restart of lab activities. Overall, these IF data on Nrf2 translocation were included to provide some mechanistic insight of the antioxidant activity of the most potent plant extracts. Therefore, the full study of Nrf2 dynamics is out of scope in the context of this study.

  1. Fig 5, the description of the HPLC-UV-DAD profiles of various plant extracts is not very useful or informative unless the authors individually interrogate the antioxidant effect of compounds corresponding with specific peaks or pooled fractions.

We thank the reviewer for rising this relevant point. The bio-guided identification of the substances generating the observed activities will be the object of future specific studies.

  1. It is not clear what is meant by "the HPLC-UV-DAD profile of the Zanthoxylum leprieurii did not show any detectable relevant peak". Not all compounds absorb light with a sufficiently high absorption coefficient to allow for UV-detection (eg saturated fatty acids). What is a peak supposed to be relevant to if it is not functionally implicated?

Thanks. According to the reviewer’s comment, the misleading word “relevant” has been deleted (line 181).

  1. Though the studies generally progress fairly logically, the link to the last set of experiments on the effect of extracts on glucose uptake seems a little tenuous. There is no reason why the effects on glucose uptake should be related to the effects of these extracts on oxidative stress alone.

Thank you for this comment. The link between oxidative stress (antioxidant properties of these plant extracts) and glucose uptake (as in-vitro measure of insulin resistance, the main pathophysiological feature of type 2 diabetes mellitus) is substantiated by a large set of experimental and clinical data (Furukawa et al. 2004, Luc et al. 2019 both cited in the paper). Moreover, some of these spices are traditionally used also in metabolic diseases (Kuate et al. Anti-inflammatory, anthropometric and lipomodulatory effects Dyglomera® (aqueous extract of Dichrostachys glomerata) in obese patients with metabolic syndrome, Functional Foods in Health and Disease 2013; 3(11):416-427). Based on these considerations, the novelty of our study is the evaluation of the interplay of these components in an experimental setting.

  1. Unclear why the effect of extracts on oxidative stress-modulated glucose uptake in HepG2 cells has such high variability and how this variability with n of only 3 still allowed for the rejection of the null hypothesis.

Thanks for this important point. Compared to basal modulation of glucose uptake by plant extracts (Table 3 and Figure 6), oxidative stress-modulated glucose uptake data actually showed a greater variability (Table 4) but still resulted significantly (p<0.05) different from glucose uptake after H2O2. As mentioned in the Statistics paragraph, the experiment shown is representative of 3 separate experiments with consistent results.

  1. Instead of superficially characterizing a multitude of extracts from different plants, it would have been more beneficial to perform a systematic in-depth characterization of selected few, and perhaps make some attempt to isolate and chemically characterize the bioactive component(s) of these extracts.

Thank you for your suggestions. The objective of this study was to identify the most promising plant extracts in the context of 11 spices, which are all associated with use in traditional medicine. Therefore, screening experiments were necessary to this purpose. Related to your helpful suggestion, we conducted more in-depth experiments for the most active plant extracts (Figures 2, 4 and 6). As already indicated for point #4, the bio-guided identification of the substances present in the most active extracts will be the object of future specific studies.

minor: non-standard abbreviations need to be spelled out on first occurence (MTS)

Thanks. “MTS” is the acronym of [3-(4,5-dimethylthiazol-2-yl)-5-(3-carboxymethoxyphenyl)-2-(4-sulfophenyl)-2H-tetrazolium]. This has been added in the Results section when was first cited (line 74).

Round 2

Reviewer 3 Report

The authors have addressed some of the concerns and/or provided reasonable explanations. The current studies could represent a good stepping stone towards more detailed characterizations.